# Self-Attention-Guided Genetic Programming: Leveraging BERT for Enhanced Tree-Structured Data Operations

## Abstract

This study investigates the application of BERT to tree-structured data which presents a significant challenge due to its lack of explicit sequential order and complex topological dependencies. While BERT has demonstrated strong performance in learning rich representations from sequential and grid-based inputs like natural language and images, its extension to non-sequential topologies remains an open research question. In this paper, we integrate BERT with genetic programming whose classic data representation is tree data structure to solve the dynamic flexible job shop scheduling (DFJSS) problem. The DFJSS problem's inherent computational complexity and highly dynamic, uncertain nature provide a rigorous testbed for our methodology. Our experiments demonstrate that BERT can effectively capture and integrate the structural information embedded in these tree-based representations. This finding highlights the versatility and adaptability of the self-attention mechanism, extending its utility beyond conventional sequential or grid-based data structures to a broader class of complex, non-sequential topologies.

## 1 Introduction

BERT (Bidirectional Encoder Representations from Transformers) is a pre-trained language representation model based on the Transformer architecture (Devlin et al., 2019). Unlike traditional left-to-right or right-to-left models, BERT leverages bidirectional self-attention to jointly condition on both left and right contexts, enabling it to capture deep contextual dependencies. Recent studies have further extended BERT beyond textual data, demonstrating its effectiveness on non-sequential data such as images (Dosovitskiy et al., 2021). These studies extend the potential BERT applications. For example, a recent study (Teixeira & Pappa, 2025) employed BERT to encode sequenced tree data, where the tree topology remained flexible while only the terminal nodes are sequenced as input to the encoder. Inspired by this idea, we extend BERT to more flexible and dynamic tree-structured data by considering the entire set of tree nodes, aiming to effectively extract latent information embedded within such flexible representations.

Genetic programming (GP) (Koza, 1994) is a hyper-heuristic framework (Burke et al., 2013; Braune et al., 2022; Planinić et al., 2021; Pillay & Qu, 2018), in which tree-based structures are among the most commonly used representations. During evolution, GP trees are iteratively modified through operations such as subtree swapping or replacement with newly generated subtrees, introducing substantial flexibility and dynamic variation. This makes the GP evolutionary process a suitable testbed for our proposed approach. To evaluate its effectiveness, we choose the DFJSS problem (Nie et al., 2013; Zhang et al., 2020), a fundamental combinatorial optimization problem with significant practical relevance in manufacturing and processing industries (Jamrus et al., 2020; Zhang et al., 2021) as the primary test case for our study. The objective of DFJSS is to determine effective schedules for processing multiple jobs on a set of machines (Hart et al., 2005), where decisions regarding machine assignment and operation sequencing must be made in dynamic environments with continuously arriving jobs (Jaklinović et al., 2021), thereby amplifying the complexity of the problem.

The goal of our proposed algorithm is to manipulate the GP tree in a more elaborate way by replacing low-contribution subtrees identified through self-attention, our proposed framework demonstrates that BERT is capable of capturing latent information that can guide effective tree manipulation.

To achieve this goal, two key challenges need to be solved:

1. Tree representation for neural models. Unlike fixed-length vectors typically used in neural networks, tree structures are hierarchical and inherently two-dimensional, making direct vectorization challenging. Recent studies (Tan et al., 2025; Zhu et al., 2025; Teixeira & Pappa, 2025; Zhang et al., 2025) propose various encoding strategies, many still struggle to preserve the full complexity of hierarchical relationships, leading to information loss. For example, Tan et al. (2025) and Zhu et al. (2025) represent GP trees using node frequency counts, which neglects the topological relationships within the tree structure.

2. Utilizing BERT's extracted information. Prior studies (Clark et al., 2019; Reif et al., 2019; Dosovitskiy et al., 2021) have shown that multi-head self-attention can capture diverse patterns in sequential (text) and grid-structured (image) data. However, how to effectively interpret and leverage these multi-head outputs remains an open question.

Our proposed framework addresses these challenges and provides empirical evidence that BERT not only learns meaningful structural patterns in tree-based data but also enables fine-grained tree manipulation through attention-guided subtree replacement.

The core assumption underlying our approach is that different heads capture different patterns, and we aggregate node-level attention scores across all heads. Nodes consistently receiving low scores across all heads are treated as trivial, allowing their corresponding subtrees to be intelligently replaced with newly generated or more meaningful structures.

The main contributions of this paper are summarized as follows:

1. Tree data vector representation: We propose a novel representation that captures the hierarchical and topological relationships within tree structure, avoiding the information loss inherent in conventional methods.

2. Attention-guided genetic operators: We develop genetic operators guided by self-attention scores extracted from BERT. By exploiting the model's ability to identify significant and trivial substructures, our approach enables a more informed and effective search.

3. Extending attention mechanisms to complex topological data: We empirically demonstrate that attention mechanisms can effectively extract structural features and dependencies from highly irregular, tree-structured data. This validates their utility in domains beyond sequential or spatial data.

## 2 RELATED WORK

GP has been widely applied to DFJSS problem, and tree-structured representation is one of most commonly used. To vectorize the tree-structure data, a classic method is to employ phenotypic characterization (PC), which encodes the observable behavior of individuals as feature vectors. The underlying assumption is that individuals with similar phenotypic vectors exhibit similar fitness values. For instance, Hildebrandt and Branke (Hildebrandt & Branke, 2015a) applied a K-Nearest Neighbors (KNN) regression model, where the Euclidean distance between PC vectors determines similarity, and the loss values of new individuals are estimated based on their nearest neighbors.

Beyond phenotypic approaches, recent work has explored genotypic(node) representations by converting GP trees into sequential forms amenable to modern machine learning techniques. Zhang et al. (Zhang et al., 2025) proposed a breadth-first search (BFS)–based encoding scheme for linearizing trees into tokenized sequences. Building on this idea, depth-first search (DFS) traversal has been suggested as a more natural alternative, yielding representations that better preserve structural interpretability.

Advances in Transformer architectures (Vaswani et al., 2023) have demonstrated the power of self-attention for extracting meaningful representations across domains from natural language processing to computer vision (Carion et al., 2020; Dosovitskiy et al., 2021). The core strength of attention lies in its ability to capture both local and global dependencies within tokenized sequences. Prior

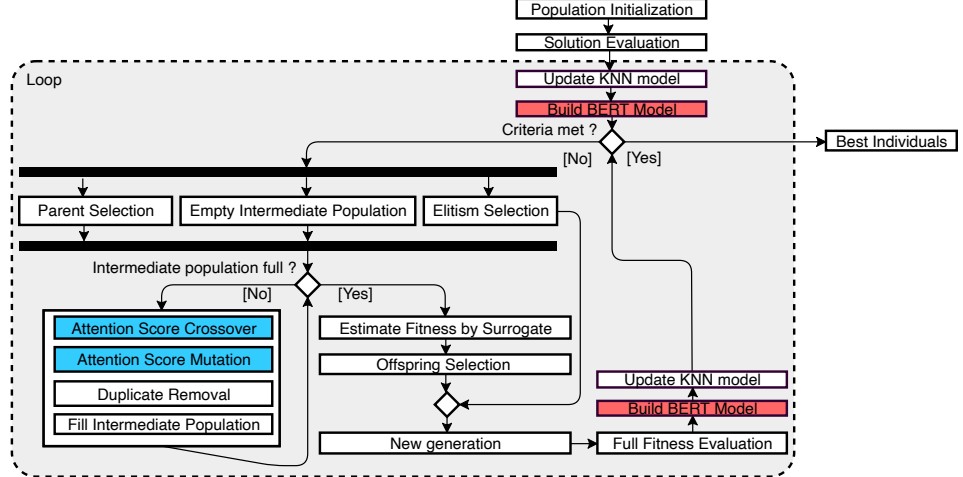

Figure 1: The flowchart of the proposed algorithm.

studies (Reif et al., 2019) have shown that attention layers in BERT encode linguistic information hierarchically: lower layers primarily reflect syntactic regularities, whereas higher layers capture semantic relationships. These findings indicate that attention mechanisms can effectively model structured relationships and extract relevant information across diverse data types.

The applications of Transformer related techniques in GP have primarily focused on symbolic regression problems (Han et al., 2025; Zhang et al., 2025), with some extensions to AutoML tasks (Teixeira & Pappa, 2025). In both symbolic regression and AutoML, the target outputs are known in advance. By contrast, the DFJSS problem is an optimization problem where the targets are unknown. As a result, relying solely on fitness values to guide the search can be inefficient.

Although Zhang (Zhang et al., 2025) proposed a method for linearizing tree-structured data using BFS, which facilitates generating replacement subtrees, this approach sacrifices some of the semantic information inherent in the tree structure.

In this paper, we instead adopt a DFS traversal to linearize the tree-structured data, resulting in sequences that better preserve semantic relationships. Furthermore, We integrate GP with the BERT self-attention mechanism to introduce an additional metric for guiding the evolutionary process. This approach proves to be more efficient than traditional methods that depend solely on random search.

## 3 PROPOSED ALGORITHM

### 3.1 AN OVERVIEW

A population of $N$ individuals is first produced as the initial generation and is subjected to a full evaluation. This evaluated population then forms the training dataset for the BERT model, as highlighted by the red box in Figure 1.

The evolutionary process proceeds in iterations. At the beginning of each iteration, a check is performed to determine if a predefined termination criterion has been met. If so, the process terminates, and the best individual from the last generation is outputted. Otherwise, the iteration continues. The previous generation undergoes GP operators for selection, self-attention score mutation and crossover(highlighted by blue box) to produce a new intermediate population. However, rather than randomly selecting the subtrees, our framework identifies the node with the lowest self-attention score and treats the subtree rooted at this node as the target for modification. Specifically, mutation replaces this subtree with a randomly generated subtree, while crossover replaces it with the subtree having the highest self-attention score from another individual. Further details are provided in Section 3.6.

Once the intermediate population is generated, a surrogate model is used to preselect the most promising individuals for the next generation.

After the new generation is composed, its individuals are fully evaluated to acquire their true fitness values, which are then used to update the surrogate and BERT model.

Finally, if the termination criterion has not been met, a new iteration begins.

During the iteration, a new BERT model is built in each iteration rather than updating the existing one for the reason: The genotypic makeup of individuals tends to converge as the iterations progress. As a result, the training parameters from earlier generations become less relevant and can negatively impact the BERT model's performance on the new, more converged population.

## 3.2 PROPOSED ARCHITECTURE

As shown in Figure 2, the proposed architecture consists of three main components: 1) the embedding(embraced by dashed box) 2) the BERT encoder 3) the global pooling and feed forward neural network. The mathematical definition is denoted as:

$$Score = GlobalAddPooling(BERT(Embeeding(\phi))) \cdot W^T \tag{1}$$

Linearize and Embed: As shown in the formula (1), we denote $\phi$ as the linearized padded tree representation. The input $\phi$ would be embedded by the process detailed in Section 3.4. By DSF, a GP tree would be linearized as a sequence, and we combine the sequence tree and routing tree of single individual into one sequence by concatenating them into one sequence. To distinguish the different linearized sequences, the segmental vector is added to indicate the different trees. The learned embedding model would be updated as the BERT model traning.

BERT Encoder: The embedded sequence would be encoded by BERT to extract the latent features by utilizing the self-attention mechanism. Since different heads capture different patterns within the sequence, we assume that a node with consistently low self-attention scores across all heads is less relevant to the other nodes. Based on this assumption, we propose an attention score–based mutation and crossover mechanism. Further details are provided in Section 3.6.

Feed Forward Neural Network: To assess an individual from semantic aspect, after the encoding, we should integrate the information extracted from encoder. To achieve this, the global adding pooling method is used. The detail is, the encoder output is firstly removed out the padding sequences, and by adding all node vectors to implement the adding pooling process. The mathematical formula is shown as $\mathbf{r} = \sum_{n=1}^{N_i} \mathbf{x}_n$. Where the $\mathbf{x}_n$ represents the $n$-th encoded token vector, and $\mathbf{r}$ is the adding pooling output vector. Finally, each individual would be processed into a vector as the input of the multi-layer of feed forward neural network(FFN). The the FFN would output the score which is used to evaluate the individuals. The reasons for employing a score, rather than a direct fitness value, are discussed in Section 3.5.

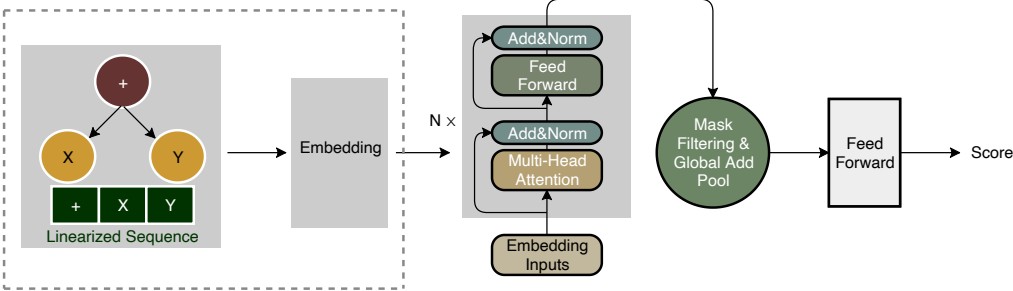

Figure 2: Neural network architecture for score generation: The input trees are first embedded using the embedding model, as illustrated in Figure 4. The resulting representations are then processed by the BERT encoder, followed by the removal of the padding mask. A global adding pooling operation is applied to obtain fixed-dimensional representations, which are subsequently passed through a FFN to produce the final scores.

### 3.3 THE SEMANTIC INFORMATION IN TREE-BASED GP INDIVIDUAL

A key fact is that a tree-based GP individual possesses an inherent semantic meaning. However, extracting this semantic content directly from its tree representation is a significant challenge. To capture this semantic information, we propose transforming the GP tree into a text sequence, as shown in Figure 3.

Specifically, by using DFS traversal, the GP tree in Figure 3 can be converted into the Polish notation format (Wikipedia contributors, 2025), such as $+(*(NIQ, PT), WINQ)$. Further, we can un-nest this expression into a simplified text sequence like $+, *, NIQ, PT, WINQ$. Finally, the linearized sequence is readable, similar to a standard sentence. Therefore, we suppose that this linearized tree-based GP representation contains valuable semantic information, which inspires us to encode it as a sequence.

Linearizing the GP tree offers several key advantages:

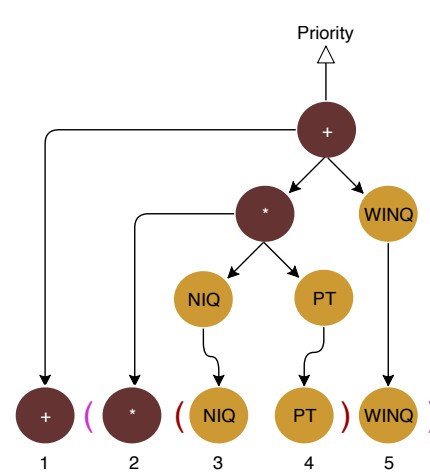

Figure 3: The two-dimensional GP tree can be linearized into a one-dimensional sequence using DFS, resulting in a Polish notation forlmula whose semantic information can be interpreted as a text sequence.

1. Comprehensive Representation: Unlike other methods, such as the frequency representation (Tan et al., 2025; Zhu et al., 2025) mentioned in a previous section, which consider only node frequencies and ignore topology, our sequence representation incorporates all nodes while the linearization mechanism preserves their topological information. This comprehensive approach ensures that critical performance-related information is retained, effectively preventing information loss.

2. Compatibility with Advanced Encoders: Text sequences are easy to manipulate and are well-suited for state-of-the-art models like the Transformer. Recent studies (Dosovitskiy et al., 2021; Carion et al., 2020) have demonstrated that the multi-head self-attention mechanism within the Transformer architecture is exceptionally good at encoding informative representations from sequential data.

In summary, by linearizing the GP tree, we leverage both of these advantages to create a representation that is not only comprehensive in its inclusion of all node information but is also highly amenable to powerful and informative encoding methods.

### 3.4 EMBEDDING LINEARIZED GP SEQUENCE

In this paper, we use two separate trees to represent the sequencing rule and the routing rule. To linearize these two trees into a single sequence, we concatenate them and use a segmental vector to identify each part. The embedding schema for this process is shown in Figure 4 where each input vector is defined as $x_i = s_i + p_i + t_i$, with $s_i$, $p_i$, and $t_i$ denoting the $i$-th segmental encoding, positional encoding, and sequence embedding vectors, respectively.

### 3.5 THE SCORE AND PAIR-WISE LOSS FUNCTION

In many studies (Zhu et al., 2025; Tan et al., 2025; Pilát & Suchopárová, 2022; Zhang et al., 2023; Hildebrandt & Branke, 2015a), the learned models are used to predict fitness values directly. However, for DFJSS problem, this approach is limited due to data instability and scarcity. Here's a breakdown of the challenges and our proposed solution.

Challenges with Direct Fitness Prediction:

1. The standard practice of estimating true fitness values for DFJSS using a limited number of instances introduces significant variance. (Hildebrandt & Branke, 2015b) For example, the average

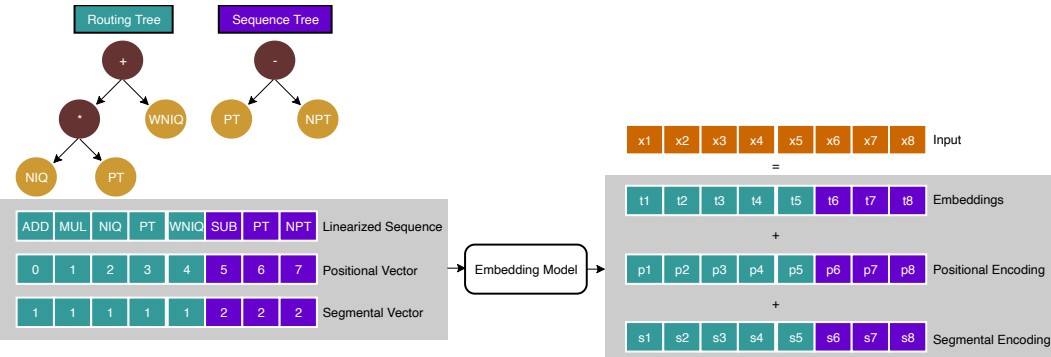

Figure 4: The process of forming a sequence from two trees and embedding this sequence as the input. This corresponds to the detailed steps within the dashed box of Figure 2

fitness over $m$ instances, the fitness values is computed by a Formula 2, can fluctuate widely. Our experiments show that this variance destabilizes the training process when employing classic loss functions such as Mean Squared Error (MSE).

$$fitness = \frac{1}{m}\sum_{i=1}^{m} CumulativeTardiness \tag{2}$$

2. Furthermore, direct fitness prediction with MSE suffers from a data augmentation limitation. In this method, each individual is treated as an independent data point with a unique fitness label. This approach ignores any relative information between individuals. In our case, the total number of individuals for training is capped at $50 \times 100 = 5000$ (50 individuals per generation over 100 generations). Using a point-wise loss function like MSE with such a small dataset can easily lead to overfitting.

To address these challenges, we propose training the learned model to output a score for each individual instead of predicting its fitness value directly. The primary objective is to align the rank of individuals based on these scores with the rank based on their true fitness values.

This approach uses a pair-wise ranking loss function, which is more robust and generalizes better than point-wise loss functions like MSE. For example, by sampling 20 data points from a dataset of 50, the number of possible unique pairs is $\binom{50}{2} = \frac{50!}{2!(50-2)!} = 1225$. This creates a much larger set of training data points compared to the original dataset size, effectively mitigating the risk of overfitting on a small dataset and making the training process more stable. The Algorithm 1 illustrates our procedure.

### 3.6 SCORE-BASED CROSSOVER AND MUTATION

In GP, crossover and mutation are used to introduce randomness and encourage exploration and exploitation. While classical GP applies these operations to random subtrees, our approach uses the self-attention mechanism to evaluate subtrees at the node level, providing a more informed guidance.

The different heads within a BERT model are designed to capture various patterns in sequential data, such as focusing on specific token types, positional relationships, or the entire sequence (Clark et al., 2019). In our work, we use 8 attention heads to capture the patterns within the linearized Polish notation formulas. Each head assigns a score to every node, indicating how much attention it receives from other nodes under that specific pattern. We then sum these 8 scores to get a total score for each node, which represents its overall importance: $N_{score} = \sum_{i=1}^{8} Score_i$, where $N_{score}$ is the total score for a node and $Score_i$ is the attention score from the $i$-th head.

Our core assumption is that a node with a very low total score is less important in all learned patterns. This allows us to make more targeted genetic operations. Instead of random replacement, we can replace the subtree rooted at a low-scoring node with a subtree from another individual that has

a high-scoring node (crossover) as shown in Figure 5, or with a new, randomly generated subtree (mutation) as shown in Figure 6. This provides a more deliberate and efficient evolutionary search.

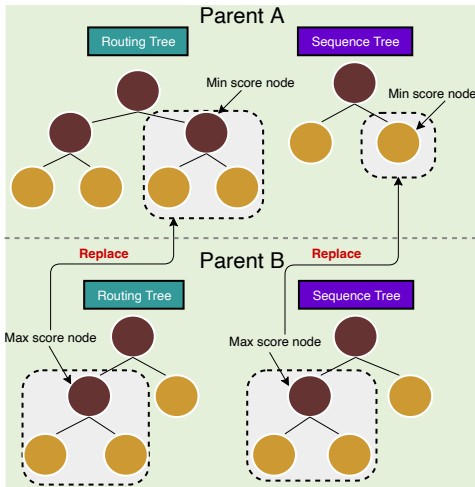

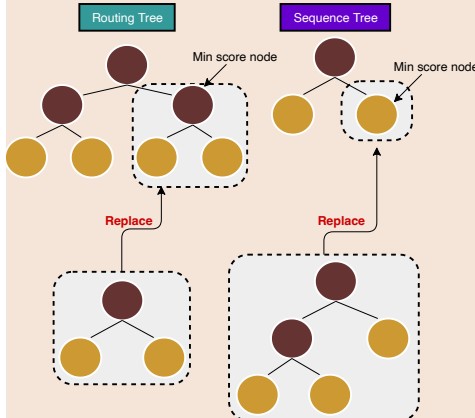

Figure 5: The crossover operation in our algorithm is guided by the attention scores derived from the BERT model. This process involves identifying the node with the minimal attention score in Parent A and the node with the maximal attention score in Parent B. The subtree rooted at the minimal-score node in Parent A is then replaced by the subtree rooted at the maximal-score node from Parent B.

Figure 6: The subtree rooted at the node with the minimal attention score is identified and then replaced with a new, randomly generated subtree. This method ensures that the mutation is not random but instead targets the least important parts of the genotype, thereby providing a more informed evolutionary search.

## 4 RESULTS AND DISCUSSIONS

### 4.1 EXPERIMENT DESIGN

**Parameter Settings:** Each individual of the algorithm is composed of terminals, as specified in Table 2, and a set of functions, namely, $\{+, -, *, /, \max, \min\}$. A protected division operator is used, where the function returns one if the divisor is zero. All programs are constrained to a maximal depth of 8. Parents are selected by tournament selection of size 3 to produce offspring. The genetic operators include crossover, mutation, and reproduction, with corresponding rates of 80%, 15%, and 5%. The algorithm terminates after a maximal number of 50 generations.

To analyze the effect of attention score-based mutation and crossover, we set up a self-attention score utilization of 0% and 80%. The 80% self-attention score utilization means 80% probability to take score-based mutation and crossover as described in Section 3.6. More experiment settings refer Appendix C.

### 4.2 OVERALL PERFORMANCE OF PROPOSED ALGORITHM

The experimental results[1] are statistically validated using Friedman's test and the Wilcoxon rank-sum test with a significance level of 0.05. This ensures that the observed differences are statistically significant and not due to random chance. In the following discussion, we denote the baseline model as KNN-GP which does not utilize self-attention scores to guide the genetic operators (mutation and crossover). The proposed model, referred to as BERT-SSGP(self-attention-score-based GP), extends the KNN-GP model by incorporating self-attention scores as guidance for the genetic operators.

---

[1]Code will be released upon acceptance of this paper.

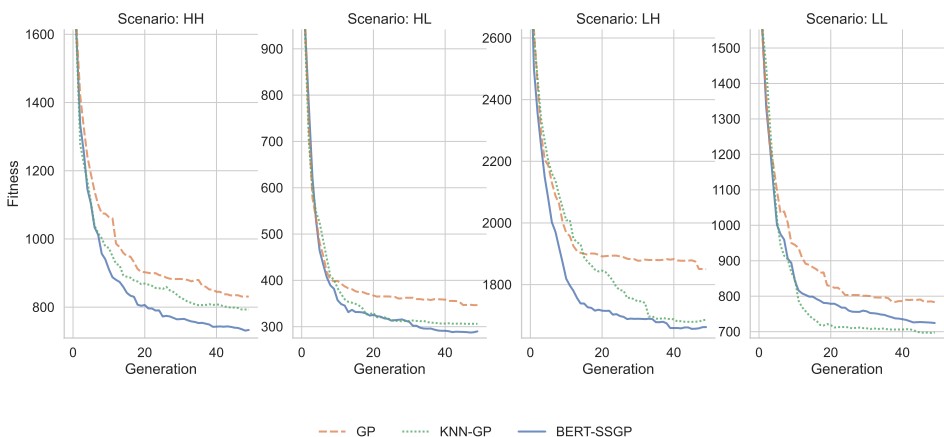

Figure 7: The curves of average fitness values according to 30 independent runs on test instances of GP, KNN-GP, BERT-SSGP with 50 generations.

Table 1: The mean (standard deviation) of objective values on test instances of GP, KNN-GP and proposed BERT-SSGP with 80% self-attentuon score utilization in 50 generations according to 30 independent runs in four scenarios.

| Algorithms | HH | HL | LH | LL | Rank |
|---|---|---|---|---|---|
| GP | 830.94(146.86)(+) | 345.61(98.47)(+) | 1846.50(331.78)(+) | 779.18(204.86)(≈) | 3.0 |
| KNN-GP | 794.20(136.84)(+) | 306.43(52.92)(+) | 1684.41(127.83)(≈) | 698.45(71.62)(≈) | 1.8 |
| BERT-SSGP | 728.91(98.57) | 290.27(57.13) | 1661.72(150.32) | 725.13(109.05) | 1.2 |

Specifically, in this model, there is an 80% probability that mutation and crossover are performed under the guidance of self-attention scores.

The results shown in Figure 7 and Table 1 demonstrate the effectiveness of the proposed algorithm.

Firstly, as shown in Figure 7, in both HH and LH scenarios, BERT-SSGP achieves a faster convergence rate than the baseline model. This indicates that, compared with random subtree selection, self-attention score based subtree selection is more effective as the self-attention scores provide informative signals that help identify inferior subtrees.

Secondly, according to Table 1, BERT-SSGP achieves the highest overall ranking among all models and outperforms the baseline model in two out of the four scenarios. However, in LH and LL scenarios, BERT-SSGP doesn't outperform the baseline because of the limitation that self-attention score mechanism could only increase the probability that the new tree generated by self-attention score guided genetic operations is better, but it can not guarantee improvement in every case. A more detailed discussion is in Appendix B

The results support the core hypothesis that a more guided, less random approach to mutation and crossover operations leads to more effective and efficient search processes within a genetic programming framework.

A key drawback of both the proposed algorithm and the KNN-GP approach is their sensitivity to the reference rules. This is a known issue, as highlighted in the paper by (Hildebrandt & Branke, 2015a). This limitation points to a need for more robust, rule-independent methods for individual evaluation and comparison.

### 4.3 PROBING INDIVIDUAL ATTENTION

The natural question that follows is, if the proposed algorithm works, what do the attention scores on each node actually signify? In this experience, we use the trained single model to compute the attention matrix on the best individual to see what information the attention matrix can express. The results shown as Figure 8, which shows 8 heads attention wights among the input sequence.

Because the sequencing rule and routing rule(embraced by red rectangle) are concatenated into a single sequence as Figure 4 shown, the attention heads within the BERT model exhibit distinct and meaningful behaviors. The experimental results reveal three primary patterns of attention:

Inter-Sequence Attention: Certain attention heads, such as Head 0, 3, and 6, predominantly attend to tokens from both sequences. This behavior suggests the model is learning the vital dependencies and interactions between the two rules, which is crucial given their collective influence on the final outcome.

Intra-Sequence Attention: In contrast, other heads, including Head 4, 5, and 7, tend to focus their attention primarily within their own sequence. This indicates that these heads are specializing in capturing the unique, inherent features and hierarchical structure of each individual rule.

Integrated Attention: A third group of heads, exemplified by Head 2, demonstrates a more balanced, integrated approach. These heads attend to both their own sequence and the other sequence simultaneously, suggesting a holistic understanding of how local features and broader inter-sequence relationships contribute to the overall representation.

This diverse range of attention patterns implies that BERT is capable of effectively identifying and learning the most critical information—both internal to each rule and in the interactions between them—within the combined tree-based sequences.

### 4.4 DOES ATTENTION SCORE CAN REALLY GUIDE THE EVOLUTION?

In previous experiments shown as the Table 1, the results from KNN-GP (with classic random subtree selection) showed an inferior performance compared to algorithms with higher self-attention score utilization. This outcome suggests that score-based mutation and crossover are effective at guiding evolution. We provide more analysis in Appendix B.

## 5 CONCLUSIONS AND FUTURE WORK

Our findings reveal that attention scores hold significant meaning within the tree structure of GP individuals. By leveraging these scores to guide genetic operations like mutation and crossover, we can effectively elaborate individuals at the genotypic level. This advancement broadens the methodological scope of GP evolution.

Furthermore, our experimental results demonstrate the efficiency of the self-attention mechanism on tree structures. We show that BERT can effectively extract both the tree's topology and the latent information embedded within its nodes.

However, our proposed approach has several clear limitations and drawbacks.

Firstly, our algorithm still relies on phenotypic duplicate removal, a process whose effectiveness is heavily dependent on the quality of the reference rules used for phenotype representation. Given our novel genotypic characterization representation, a promising solution is to implement a genotype-level similarity measure to assess the similarity between two individuals, providing a more robust duplicate removal method.

Secondly, our model suffers from a significant increase in training time compared to classic phenotype-based models. This drawback is particularly pronounced with BERT, as its attention score computation complexity is $O(n^2)$, where $n$ is the sequence length. Given that the linearized sequence can be quite long, the computational cost is substantially higher than that of traditional machine learning methods.

Finally, we identify a key direction for future research. While our work shows that attention scores can guide evolution, a more advanced step is to use this insight for generative purposes. If the attention score indicates that a specific subtree is inferior, the next logical step would be to develop a mechanism to generate a superior subtree in a more deliberate and fine-grained way, rather than simply replacing the inferior one with a randomly generated substitute.

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

## A  DETAILS OF PAIR-WISE LOSS FUNCTION

Our methodology begins by grouping individuals based on their fitness values. As the evolutionary process progresses and the population converges, individuals often achieve identical fitness scores. These individuals are partitioned into distinct groups, and we subsequently compute the score variance within each group. The underlying principle is that a well-calibrated predictive model should assign scores with minimal variance to individuals that share the same fitness value. This approach provides a measure of the model's consistency and predictive reliability for equally-performing individuals.

---

**Algorithm 1** Pairwise Loss Function

---

**Input:** scores, labels, $\lambda_{\text{var}}$
**Output:** loss value $\mathbb{L}$
1: $\mathbb{L} \leftarrow 0$
2: variation $\leftarrow 0$
3: $n \leftarrow$ length of labels
4: mask $\leftarrow$ list of $n$ zeros
5: unique score list $\leftarrow []$
6: groups $\leftarrow$ unique elements in labels
7: **for** each $g \in$ groups **do**
8:     indices $\leftarrow \{i \mid label_i = g\}$
9:     values $\leftarrow \{score_i \mid i \in$ indices$\}$
10:     $\mathbb{L} \leftarrow \mathbb{L} +$ Variance(values)
11: **end for**
12: $\mathbb{L} \leftarrow \mathbb{L} \cdot \lambda_{\text{var}}$
13: **for** $i = 0 \ldots n - 1$ **do**
14:     **for** $j = i \ldots n - 1$ **do**
15:         **if** labels[$i$] = labels[$j$] **then**
16:             $\mathbb{L} \leftarrow \mathbb{L} + |$scores[$i$] $-$ scores[$j$]$|$
17:         **else**
18:             **if** labels[$i$] > labels[$j$] **then**
19:                 sign $\leftarrow -1$
20:             **else**
21:                 sign $\leftarrow 0$
22:             **end if**
23:             $\mathbb{L} \leftarrow \text{ReLU}\big(\mathbb{L} + ($scores[$i$] $-$ scores[$j$]$) \cdot$ sign$\big)$
24:         **end if**
25:     **end for**
26: **end for**
27: **return** $\mathbb{L}/(n \cdot (n-1)/2) +$ variance

---

Following the variance-based analysis, we employ a pairwise ranking loss function to compute the overall loss value. The core premise of this function is to enforce a consistent relationship between the predicted scores and the actual fitness ranks. If two individuals possess the same fitness value, the loss function is designed to penalize any discrepancy in their predicted scores. Conversely, if there is a rank mismatch—where the predicted rank does not align with the true fitness-based rank—the difference between their scores is added to the total loss. This mechanism directly guides the model to learn the correct ordinal relationships among individuals.

To ensure that only ranking errors contribute to the loss, we apply the Rectified Linear Unit (ReLU) function to the computed loss values. This function ensures that if the predicted rank is correct (i.e., the pairwise difference is non-positive), the resulting loss is zero. If the rank is incorrect, the loss value remains positive, thereby penalizing the model for misordering. The complete process is formally detailed in Algorithm 1.

## B  A DEMONSTRATION OF ATTENTION SCORE

In 4.4, the experiment results show that the attention score can guide the evolution. In this section, we would discuss more details and propose an explanation.

We propose a hypothesis to explain this: by targeting and replacing nodes with the lowest average scores, we can potentially increase the lower bound of an individual's overall score. While this mechanism doesn't guarantee an improvement with every operation (because new subtrees are generated randomly), it increases the probability of improvement.

Let's formally represent this process to understand the potential for improving an individual's performance.

Represent the entire tree structure as $T$, which consists of $N$ nodes, $\{v_1, v_2, \ldots, v_N\}$. Each node $v_i$ has a score, $s(v_i)$. The overall performance of the tree, $E(T)$, is defined by its minimum node score as $E(T) = \min_{i=1}^{N} s(v_i)$.

Let $v_{\min}$ be the node with the lowest score in tree $T$, so $v_{\min} = \arg\min(s(v_i)), i \in \{1, 2, \ldots, n\}$, and its minimum score is $s_{\min} = s(v_{\min})$.

In a genetic operation, we replace the subtree rooted at $v_{\min}$ with a new, randomly generated subtree, $T_{new}$. This new subtree contains $m$ nodes, $\{u_1, u_2, \ldots, u_m\}$, each with a score $s(u_j), j \in \{1, 2, \ldots, m\}$.

The new tree, $T'$(the new tree after the replacement), has a minimum score of:

$$E(T') = \min(\{s(v_i) \mid v_i \in T, v_i \neq v_{\min}\} \cup \{s(u_j) \mid u_j \in T_{new}\})$$

If all node scores in the new subtree, $T_{new}$, are greater than or equal to the original minimum score, $s(u_j) \geq s_{\min}$ for all $j$, then the new tree's minimum score will be at least as good as the original, i.e., $E(T') \geq s_{\min}$. This is the ideal scenario where the operation is guaranteed to improve or maintain the tree's lowest score.

However, since the new subtree is generated randomly, it might contain nodes with scores lower than the original minimum. In this case, the new tree's minimum score, $E(T')$, could be lower than $E(T)$. But by targeting the original lowest-scoring node for replacement, we are increasing the probability of generating a new subtree with a higher minimum score, thereby raising the overall lower bound of the individual. This process offers a probabilistic mechanism for improving the tree's overall performance.

From the above derivation, the most significant step is to increase the probability that the randomly generated subtree is better than the original replaced subtree.

## C  EXPERIMENT DESIGN

**Dataset:**  To measure the performance of proposed algorithm, four scenarios are considered based on three key factors.

- 1) Expected job arrival rate / system utilization level:

  The system utilization level is directly relative with the job arrival rate, and we denote $E(u)$ as the expected utilization level, $\mathbb{E}(t)$ as the expected processing time of all operations on all machines, and $\mathbb{E}(in)$ as the expected time interval of job arrivals. Mathematical expression as:

  $$E(u) = \frac{w\mathbb{E}(t)}{k\mathbb{E}(in)}$$

  where $k$ represents the number of machines, and $k$ represents the number of workcenter.

  In this paper, we set the $E(u)$ 0.9 to all four scenarios to simulate a busy factory. The arrival intervals follow the exponential distribution, namely $X(in) \sim Exp(\mathbb{E}(in))$.

- 2) Heterogeneity of the processing time:

  The processing time of any job $J_j$'s operation $O_{j,i}$ on machine $M_m$ $t_{j,i,m}^{pro}$ follows the uniform distribution $U[L_p, H_p]$, that is $t_{j,i,m}^{pro} \sim U[L_p, H_p]$, where the $L_p$ and $H_p$ denote

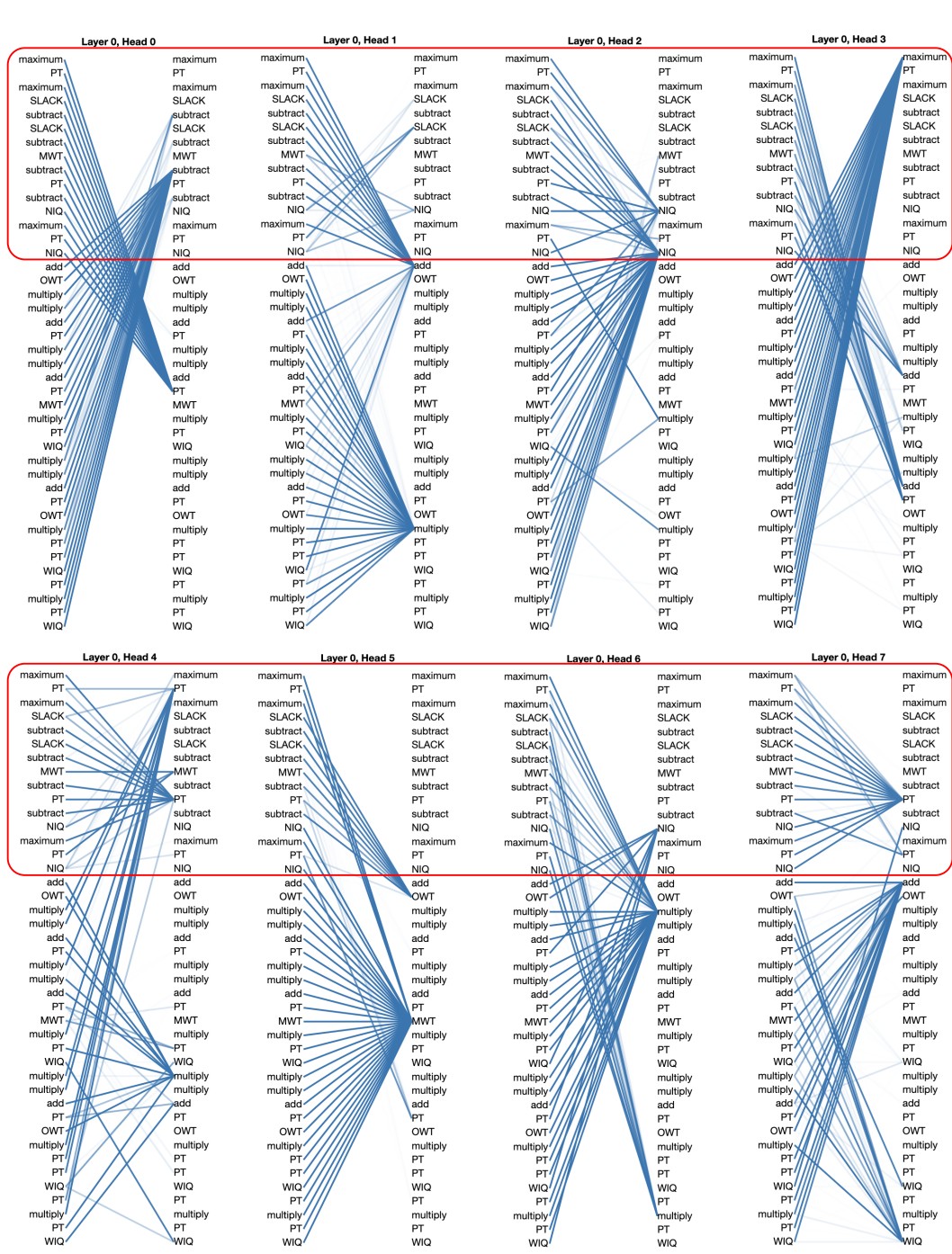

Figure 8: BERT attention heads, In the example attention maps, the darkness of a line indicates the strength of the attention weight.

Table 2: The terminal and function sets.

| PART | DESCRIPTION |
|---|---|
| **Machine-related** | |
| NIQ | The number of operations in the queue |
| WIQ | Current work in the queue |
| MWT | Waiting time of a machine |
| **Operation-related** | |
| PT | Processing time of an operation |
| NPT | Median processing time for next operation |
| OWT | Waiting time of an operation |
| **Job-related** | |
| WKR | Median amount of work remaining of a job |
| NOR | The number of operations remaining of a job |
| SLACK | The slack of the job $J_i$ at time $t$ |
| TIS | Time in system |

the lower and upper bounds of processing time respectively. We define processing time $t_{j,i,m}^{pro} \sim U[5, 15]$ as the high heterogeneity processing time, and processing time $t_{j,i,m}^{pro} \sim U[10, 20]$ as the low heterogeneity processing time.

- 3) Due date tightness:
  As we denote the due date of a given job $J_j$ as $t_j^{due}$, we denote a due date factor $\alpha_j \sim U[L_d, H_d]$. In this paper, we categorize the due date factor $\alpha_j$ into two types of tightness range: the high tension of due date with $\alpha_j \sim U[1, 2]$ and the low tension of due date with $\alpha_j \sim U[1, 3]$. The due date $t_j^{due}$ mathematical calculation is:

$$t_j^{due} = t_j^{arr} + \alpha_j \sum_{i=1}^{q_j} \left( \frac{\sum_{m=1}^{k_{j,i}} t_{j,i,m}^{pro}}{k_{j,i}} \right), \alpha_j \sim U[L_d, H_d]$$

  The formula calculates the due time ($t_j^{due}$) for job $J_j$ as its arrival time ($t_j^{arr}$) plus a randomly generated duration. This duration is determined by a scaling factor $\alpha_j$, which is drawn from a uniform distribution between $L_d$ and $H_d$, multiplied by the sum of the average processing times for each operation of the job. The term $\frac{\sum_{m=1}^{k_{j,i}} t_{j,i,m}^{pro}}{k_{j,i}}$ represents the average processing time of operation $O_{j,i}$ across all $k_{j,i}$ available machines.

Based on these descriptions, the four scenarios are:

- 1) $HH$: the high heterogeneity of processing time $t_{j,i,m}^{pro} \sim U[5, 15]$ and high tension of due date $\alpha_j \sim U[1, 2]$
- 2) $HL$: the high heterogeneity of processing time $t_{j,i,m}^{pro} \sim U[5, 15]$ and low tension of due date $\alpha_j \sim U[1, 3]$
- 3) $LH$: the low heterogeneity of processing time $t_{j,i,m}^{pro} \sim U[10, 20]$ and high tension of due date $\alpha_j \sim U[1, 2]$
- 4) $LL$: the low heterogeneity of processing time $t_{j,i,m}^{pro} \sim U[10, 20]$ and low tension of due date $\alpha_j \sim U[1, 3]$

And in our experiments, we set up three workcenters and each workcenter has two machines.

## D  DYNAMIC JOB FLEX SHOP SCHEDULING

In DFJSS problem, $m$ machines denote by $\mathcal{M} = \{M_1, M_2, ..., M_m\}$, and each machine belongs to workcenters denoted as $\mathcal{W} = \{W_1, W_2 ..., W_w\}$. $n$ jobs $\mathcal{J} = \{J_1, J_2, ..., J_n\}$ would arrive

Table 3: The parameter settings of the proposed method.

| PARAMETER | VALUE |
|---|---|
| Population size | 50 |
| Number of generations | 100 |
| Number of instances per generation | 2 |
| Method for initialising population | Ramped-half-and-half |
| Initial minimum/maximum depth | 1 / 6 |
| Elitism | 10 |
| Maximal depth | 8 |
| Crossover rate | 0.80 |
| Mutation rate | 0.15 |
| Reproduction rate | 0.05 |
| Terminal/non-terminal selection rate | 10% / 90% |
| Radius $\delta$/capacity $\kappa$ | [0, 1, 2, 3, 4, 5] / 1 |
| Parent selection | Size-3 tournament selection |
| Self-attention score utilization | 0 / 0.8 |

dynamically, and each job $J_j$ has a sequence of operations $\mathcal{O}_j = \{O_{j1}, O_{j2}, ..., O_{jl_j}\}$ that need to be processed one by one, where $l_j$ is the number of operations of job $J_j$. Each completed job $J_j$ has a completed time denoted as $t_j^{con}$ and a due time denoted as $t_j^{due}$. $tard(J_j) = max\{t_j^{con} - t_j^{due}, 0\}$ is denoted as the tardiness which is lower bounded by 0, that means if the completed time is earlier than due time, the tardiness is 0.

Each operation $O_{ji}$ can be processed on more than one machine $M(O_{ji}) \subseteq \pi(O_{ji})$ (**?**). Thus, the machine that processes an operation determines its processing time $\delta(O_{ji}, M(O_{ji}))$.

Below are the main constraints of the DFJSS problems:

- 1. A machine can only process one operation at a time. For any given $M_k$:

$$t_{jik}^{start} \notin [t_{mnk}^{start}, t_{mnk}^{con}], \forall j, i \neq m, n \tag{3}$$

  where $t_{jik}^{start}$ represents the star time of operation $O_{ji}$ on machine $M_k$, and $t_{mnk}^{start}$, $t_{mnk}^{con}$ represent the start time and the completion time of operation $O_{mn}$ on machine $M_k$ respectively.

- 2. Each operation can be handled by only one of its candidate machines.

$$\sum_{k \in M(O_{ji})} x_{jik} = 1, \forall j, i > 1, x_{jik} \in \{0, 1\} \tag{4}$$

  where $x_{jik} = 1$ if operation $O_{ji}$ is assigned to machine $M_k$, otherwise, $x_{jik} = 0$.

- 3. An operation cannot be handled until its precedents have been processed. For any operation $O_{ji}$ and it's preceding $O_{j(i-1)}$

$$t_{j,i}^{start} > t_{j(i-1)}^{con}, \forall j, i > 1 \tag{5}$$

  where $t_{j,i}^{start}$ represents the start time of $O_{ji}$ and $t_{j(i-1)}^{con}$ represents the completion time of the preceding operation $O_{j(i-1)}$

- 4. Once started, the processing of an operation cannot be stopped.

In this paper, we use the cumulative tardiness of whole evaluation process as the only measurement:

$$CumulativeTardiness = \sum_{j=1}^{n} tard(J_j), tard(J_j) = max\{t_j^{con} - t_j^{due}, 0\} \tag{6}$$

where $tard(J_j)$ represents the tardiness time of a job $J_j$. The objective of GP solution is to minimize the $CumulativeTardiness$.