# OpenReview forum: "Self-Attention-Guided Genetic Programming: Leveraging BERT for Enhanced Tree-Structured Data Operations"
_ICLR.cc/2026/Conference — ICLR 2026 Conference Withdrawn Submission_

### Official Review · Reviewer_szRp · 2025-10-30

**Soundness:** 1
**Presentation:** 1
**Contribution:** 2
**Rating:** 2
**Confidence:** 3

**Summary:**

The paper investigates how self-attention from a Transformer model (BERT) can be used to steer GP. Individuals in GP are represented as trees and are first converted into a token sequence in prefix notation through depth-first traversal. This sequence is then passed to a BERT encoder, which computes self-attention scores for each token corresponding to a tree node. These attention scores are subsequently used to influence the genetic operators: subtrees at nodes with low attention scores are selected for mutation, while subtrees at nodes with high attention scores from other individuals are selected for crossover. During the evolutionary process, the model is trained iteratively using a pairwise ranking loss to approximate the relative fitness ranking of individuals. The approach is applied to the DFJSSP, where GP is used to generate dispatching rules, and it is compared against standard GP and a surrogate-based baseline.
According to the authors, the contributions of the paper are:
- Linearizing GP trees into token sequences using depth-first traversal (prefix/Polish notation), enabling the application of Transformer/BERT models to tree-structured GP individuals.
- Using self-attention scores produced by the BERT encoder to guide genetic operators: subtrees with the lowest attention scores are selected for mutation, while subtrees with the highest attention scores from other individuals are selected for crossover.
- Integrating a BERT-based surrogate model that is trained iteratively with a pairwise ranking loss to produce an estimated score for each individual, which then guides the evolutionary search instead of relying solely on true fitness evaluations.

**Strengths:**

- The paper introduces a novel combination of genetic programming and Transformer-based self-attention. Using BERT to evaluate GP individuals and to guide mutation and crossover via attention scores represents a creative hybrid of neural and symbolic methods.
- The approach is implemented end-to-end, including tree linearization, sequence encoding, surrogate training, and attention-guided operator selection. Experiments compare the method against baseline GP variants and show some performance improvements.
- The work contributes to the broader direction of neural-assisted evolutionary search and may inspire further research on integrating pretrained language models into symbolic optimization.

**Weaknesses:**

- The main claim is not demonstrated. There is no quantitative analysis of attention scores, and no ablation comparing attention-guided vs. random subtree selection to isolate the effect of attention. In addition, evaluating only against random selection is insufficient: there are established alternative node selection strategies in GP (such as depth-biased or semantic-aware mutation), which would serve as stronger and more informative baselines than random replacement.

- Converting GP trees into prefix notation via DFS is not new. This representation has been used in earlier forms of linear GP, yet the the authors do not relate their approach to prior work or explain why a Transformer should be preferred over models explicitly designed for structured data.

- Key design choices (such as population size, limiting training samples to 5000 individuals, and the specifics of the node replacement strategy) are not motivated. Baselines are restricted to classical GP and KNN-GP..

- Important components are described too briefly, including model configuration, hyperparameters, and operator selection rules, which makes the approach difficult to reproduce.

- The paper mixes motivation, method description, and implementation details, and terminology is inconsistent. Some figures are not sufficiently explained, and language issues reduce clarity and make several sections difficult to follow.

**Questions:**

- How do you validate that self-attention scores meaningfully reflect the importance of subtrees?
- Did you perform any quantitative analysis (e.g., correlation between attention scores and subtree fitness contribution)?
- Why is the evaluation limited to attention-guided vs. random subtree selection? Have you considered comparing against established GP node-selection strategies?
- Tree linearization via prefix notation (DFS / Polish notation) has been used in Gene Expression Programming and variants of Linear GP. Can you clarify what is novel about your representation and how it differs from these prior approaches?
- Why did you choose a Transformer over models explicitly designed for tree or graph structure? What advantages do you expect from Transformers in this setting?
- Several design choices (population size, capping training samples at 5,000 individuals, node replacement strategy) seem arbitrary. How were these parameters selected, and did you explore alternatives?
- Why are baselines limited to classical GP and KNN-GP?
- Could you provide more detail about the surrogate model configuration (hyperparameters, training schedule, selection of attention heads)?
- Some figures and terminology are not clearly explained, and several sentences are difficult to follow. Are you planning to provide an improved version with clearer diagrams, consistent terminology, and language polishing?

---

### Official Review · Reviewer_hXRT · 2025-10-31

**Soundness:** 2
**Presentation:** 1
**Contribution:** 3
**Rating:** 2
**Confidence:** 3

**Summary:**

This paper introduces a method that integrates BERT with genetic programming (GP) to solve the dynamic flexible job shop scheduling (DFJSS) problem. The core contribution is adapting BERT to tree-based structures and using attention scores to intelligently guide the crossover and mutation operators.

**Strengths:**

The paper uses BERT's self-attention scores to guide the genetic search operators. In general, adaptively selecting mutation and crossover operators to guide the search process is a very promising approach. I appreciate the authors for evaluating the method on complex DFJSP.

**Weaknesses:**

- The key weakness is the potential computational overhead that retraining the BERT model in every generation can cause. This is likely orders of magnitude slower than the baselines.
- The paper does not report any runtime performance to assess practical feasibility.
- The experimental setup can be improved. It is difficult to read the paper without consulting the appendix multiple times. Those four scenario should be part of the main paper.
-The performance gains appear inconsistent. the proposed BERT-SSGP fail to outperform the KNN-GP baseline in two of the four scenarios.
- This work is closely related to the field of adaptive parameter tuning where using deep reinforcement learning (DRL) has been studied extensively to tune the parameters of evolutionary algorithms; however, I miss references to any of such papers.
- The paper cites Wikipedia to refer to Polish notation. Is there no credible academic reference to cite it properly?

**Questions:**

- In Table 1, BERT-SSGP does not show a statistically significant improvement over KNN-GP in the LH and LL scenarios. Can you your thoughts on why the attention-guidance mechanism is less effective in these specific settings?
- Studies on DJSP typically adapt FJSP benchmarks for evaluation. Would it be possible to evaluate on those benchmark (or generated) datasets publicaly made available by other studies for instance https://doi.org/10.1016/j.asoc.2025.114008?
- What is the motivation for using BERT beyond that it works well for images and text? I see a sequential decision-making paradigm like DRL as a better fit for guiding the evolutionary search process. What are the authors' thoughts on this alternative approach, and why did they choose BERT over more naturally sequential methods like reinforcement learning for this optimization problem?

---

### Official Review · Reviewer_AGgt · 2025-11-01

**Soundness:** 2
**Presentation:** 2
**Contribution:** 2
**Rating:** 2
**Confidence:** 4

**Summary:**

The paper proposes to use a Bert encoder and Multi-Head Attention in order to evaluate subtrees for Genetic Programming. The results are slightly better than vanilla GP at the cost of increased time for inference on the subtrees.

**Strengths:**

The paper proposes an original method to improve GP.
The results are slightly better than vanilla GP.

**Weaknesses:**

The approach is much more complicated than usual GP and it improves only slightly on GP.

**Questions:**

Can you detail how the BERT model is trained?

---

### Official Review · Reviewer_7dyi · 2025-11-04

**Soundness:** 2
**Presentation:** 3
**Contribution:** 2
**Rating:** 4
**Confidence:** 4

**Summary:**

This study introduces a novel framework that integrates BERT with genetic programming to enhance tree-structured data operations by linearizing GP trees into sequences using depth-first search, employing BERT's self-attention mechanism to identify significant nodes, and guiding genetic operations through attention scores—where mutation replaces low-scoring subtrees with random ones and crossover substitutes them with high-scoring subtrees from other individuals, while using a pairwise ranking loss for robust evaluation. The method demonstrates superior convergence and effectiveness in dynamic flexible job shop scheduling problems compared to baseline approaches.

**Strengths:**

The paper introduces a novel approach that combines BERT with genetic programming to solve the dynamic flexible job shop scheduling problem. By leveraging self-attention mechanisms, the model effectively captures structural information in tree-based representations, improving the efficiency of genetic operations like mutation and crossover. The approach demonstrates the potential of extending BERT’s self-attention beyond sequential data to complex tree structures, offering a more efficient search process.

**Weaknesses:**

The paper does not provide an in-depth analysis of the computational complexity introduced by the use of BERT. The quadratic complexity of O(n^2) in attention score computation could make the method impractical for larger DFJSS instances, where long sequences significantly increase training time. Although the paper briefly acknowledges this issue, it does not offer a solution or discuss ways to mitigate this computational burden. Second, while the paper claims that self-attention scores can effectively guide genetic operations, the justification for this approach is weak. It is unclear how attention scores translate into meaningful guidance for genetic operators. Lastly, the experimental design in the paper is highly limited, as it only compares the proposed method with two older baseline approaches (from 1994 and 2014), which may not represent the current state-of-the-art methods in the field. Due to the lack of comparisons with more modern baselines and the absence of ablation studies, it is difficult to fully assess the advantages of the proposed method and to clearly differentiate the contributions of individual modules (such as the BERT model and attention-guided genetic operators) to the overall performance. Additionally, the scalability of the method to larger, more practical datasets, as well as its ability to generalize beyond DFJSS, has not been sufficiently addressed.

**Questions:**

1. The paper does not address the computational burden caused by BERT’s quadratic time complexity, especially for larger DFJSS instances with long sequences. Please discuss in detail the impact of this computational burden, and provide time-related experimental data if necessary.
2. While self-attention scores are claimed to guide genetic operations, it remains unclear how these scores specifically influence genetic operators. A clearer explanation of this mechanism across various problem instances is needed.
3. The paper compares the proposed method only with two older baselines. The method should be tested against more modern state-of-the-art scheduling methods to better validate its advantages and provide a performance analysis, including time-related metrics.
4. The paper does not sufficiently explore the scalability of the approach to larger, practical datasets, nor its ability to generalize beyond DFJSS. Further evaluation in a broader range of application scenarios is needed.

---

### Note · Authors · 2025-11-15

I have read and agree with the venue's withdrawal policy on behalf of myself and my co-authors.